# Intelligent Automatic Segmentation of Wrist Ganglion Cysts Using DBSCAN and Fuzzy C-Means

**DOI:** 10.3390/diagnostics11122329

**Published:** 2021-12-10

**Authors:** Kwang Baek Kim, Doo Heon Song, Hyun Jun Park

**Affiliations:** 1Department of Artificial Intelligence, Silla University, Busan 46958, Korea; 2Department of Computer Games, Yong-In Art and Science University, Yongin 17145, Korea; dsong@yasu.ac.kr; 3Division of Software Convergence, Cheongju University, Cheongju 28503, Korea; hyunjun@cju.ac.kr

**Keywords:** ganglion cyst, fuzzy C-means, DBSCAN, machine learning, pixel clustering

## Abstract

Ganglion cysts are common soft tissue masses of the hand and wrist, and small size cysts are often hypoechoic. Thus, identifying them from ultrasonography is not an easy problem. In this paper, we propose an automatic segmentation method using two artificial intelligence algorithms in sequence. A density based unsupervised learning algorithm called DBSCAN is performed as a front-end and its result determines the number of clusters used in the Fuzzy C-Means (FCM) clustering algorithm for quantification of ganglion cyst object. In an experiment using 120 images, the proposed method shows a higher extraction rate (89.2%) and lower false positive rate compared with FCM when the ground truth is set as the human expert’s decision. Such human-like behavior is more apparent when the size of the ganglion cyst is small that the quality of ultrasonography is often not very high. With this fully automatic segmentation method, the operator subjectivity that is highly dependent on the experience of the ultrasound examiner can be mitigated with high reliability.

## 1. Introduction

Ganglion cysts are common benign soft tissue tumor. More than half of them occur in the wrist. They are composed of mucoid material surrounded by fibrous tissue without a synovial lining [1]. Ganglion cysts are often painless, but appropriate treatment is required when the patient feels pain, stiffness, or interfering with joint movement. The main treatment options are either aspiration that drains the fluid out of the cyst with a needle and syringe or surgical excision. Open surgical excursion is known to have lower recurrence rate, but it has higher rates of complication and longer recovery period [2,3,4]. Statistically, ganglion cysts are found more in women than men, and are more common in the age group 20–40 than other age groups [5].

Ultrasonography is used to identify cysts due to its ability to differentiate a solid mass from a cyst [6]. The most common location where ganglion cysts occur is the dorsal component of the scapholunate ligament, followed by volar between the radial artery and flexor carpi radialis [7]. The shape of ganglion cysts of the wrist from ultrasonography can be described as anechoic, oval, round, or lobulated [6]. In treatment phase, ultrasonography also gives guidance for aspiration and injection and the information of the exact location of the pedicle for surgery [8].

However, the common problem against using ultrasonography in diagnosis is its well-known examiner subjectivity [9,10]. It was reported that small wrist ganglion cysts (≤10 mm in the mean largest dimension) often appeared hypoechoic without posterior acoustic enhancement and did not fulfill the criteria for a simple cyst thus examiner’s experience is one of key factors for correct observation [6,7]. An automatic segmentation approach using intelligent computer vision techniques can mitigate such subjectivity in the image analysis [11]. Automatic segmentation in ultrasound images, however, can be difficult for numerous reasons such as insufficient contrast and resolution of image, speckle noise, which is inherent property of ultrasound imaging modality [12]. For the ganglion cyst segmentation problem, a computerized cyst measuring system with different modality- functional infrared (fNIR) spectroscopy was introduced [13], as well as automatic retinal cyst detector by optical coherence tomography [14]. However, those works are not appropriate for segmenting ganglion cyst with ultrasonographic images in fully automatic approach (such as the approach proposed in this paper).

Previously, a pixel clustering algorithm called Fuzzy C-Means (FCM) [15] as used to detect ganglion cysts automatically [16]. FCM is a popular unsupervised learning algorithm that assigns each data a degree of fuzzy membership with the distance measure to the nearest cluster. FCM allows each data (pixel in the image in this problem) to belong to two or more clusters depending on the degree of membership to each cluster. For n data vectors and c fuzzy clusters (c < n), FCM classifies the image into clusters having similar pixels in the feature space with iteratively minimizing the cost function dependent on the distance of the pixels to the cluster centers in the feature domain. With such flexibility, FCM has been successful in segmentation problems in many medical and engineering domains [17,18,19,20,21,22,23].

However, FCM is often criticized due to two aspects of the original algorithm, namely noise sensitivity and static initialization of algorithm parameters (that is, the number of clusters and the initial membership matrix). Some other types of FCM are introduced to resolve the noise sensitivity problem [24,25,26,27] by considering the relationship of neighbor pixels but we are interested in the other side of improving FCM functionality in ganglion cysts image segmentation problem (that is, to determine the number of clusters in FCM dynamically).

There have been several approaches in this line of FCM improvement as well. Histogram analysis is carried out for counting peaks of the given image, and the resulting number is given to the number of clusters for FCM [17,28]. While it is a reasonable alternative to make the FCM adaptive to the input image, the number of peaks in the histogram is not necessarily connected to the actual number of clusters. A cooperative learning approach that uses fuzzy ART that does not need hard initialization of the number of clusters was successful in inflamed appendix segmentation problem [29]. In this paradigm, fuzzy ART works as a front end of FCM and its result is used to determine the number of clusters initialization for FCM. Recently, a density based FCM is proposed to control the initializations dynamically by computing the density of the data pixels within FCM algorithm. The result was plausible, but the improvement of that new D-FCM was not significant and it is doubtful that it was statistically significantly better than FCM for the segmentation problem [18].

In this paper, we propose another cooperative pixel clustering method to locate ganglion cysts in the wrist area automatically from ultrasonography with FCM as the back end. Density-based spatial clustering of applications with noise (DBSCAN) [30] is used as the front end to determine the number of clusters for FCM. The advantage of DBSCAN algorithm is that it can find different shapes and sizes of clusters as opposed to FCM that finds circular or oval clusters. Thus, if it works, the shape of the cluster would be more natural than those of FCM extractions. After DBSCAN is applied, the number of clusters for FCM is computed by the equation that will be explained in Section 2.2 and FCM is applied to finish the segmentation process. DBSCAN has been used in medical image segmentation [31] but it is also well known that DBSCAN is weak when the border objects of two clusters are relatively close [32]. Although main algorithms used in this paper is based on standard DBSCAN and FCM algorithms as well as several image processing algorithms used in our method, they are implemented by us in C# under Microsoft Visual Studio 2019.

## 2. Materials and Methods

### 2.1. Preprocessing of the Image Data

Very often, the area of target organ is too dark and has low intensity contrast from the background. Thus, an image enhancement technique is applied in the preprocessing phase. We use fuzzy stretching since it is effective when the intensity distribution is not uniform in the input image so that the threshold selection process itself has a high fuzziness [33]. We use trapezoidal membership function that is simple and effective in image enhancement applications where the intensity distribution seems to be irregular [34], as shown in Figure 1.

For given intensity value *G*, the membership degree *μ*(*G*) is defined as following.
(1)$$\mu \left(G\right)=\{\begin{array}{c} 0,\\ \frac{G-I_{min}}{I_{midL}-I_{min}},\\ 1,\\ \frac{I_{max}-G}{I_{max}-I_{minR}}, \end{array}\begin{array}{c} G\leq I_{min}\;or\;I_{max}\leq G\\ I_{min}<G<I_{midL}\\ I_{midL}<G<I_{midR}\\ I_{midR}<G<I_{max} \end{array}$$
(2)$$\begin{array}{c} \delta =\frac{\left(I_{mid}+\left(I_{max}+I_{min}\right)/2\right)}{255}\\ \alpha -cut=\left(\delta +0.5\right)\wedge 1 \end{array}$$

In Equation (2), *I_max_* and *I_min_* denote the maximum and minimum intensity value of the input image and *α*-cut is the minimum intensity threshold that the stretching occurs and ∧ is the fuzzy Min operator.

After fuzzy stretching as shown in Figure 2b, we remove dark part of the image by filtering as shown in Figure 2c. This yellow color represented noise area is considered in the final object forming process after FCM quantization as shown at the end of Section 2.

The effect of fuzzy stretching is as shown in Figure 2.

### 2.2. DBSCAN as the Front-End Segmentation Process

DBSCAN has two parameters—*Pts^min^* that means the threshold for the number of neighbors and radius *ε*. Objects with more than *Pts^min^* neighbors within this radius ε are core points and DBSCAN tries to find partitions that satisfy this minimum density and are separated by areas of lower density points [32]. All neighbors within the *ε* radius of a core point are part of the same cluster and neighborhoods of any core point in that cluster are transitively included to that cluster. Thus, the number of clusters is adaptively determined and DBSCAN can generate clusters of any arbitrary shape. In this specific experiment, we set *Pts^min^* = 3, *ε* = 1.

In this paper, we take histogram analysis to the input image first and divide two areas with respect to the average intensity. For each pixel in the region of interest, kth intensity *Da_k_* belongs to in DBSCAN_ct_ if there exists the same property (high or low density) of continuous series of intensities as shown in Figure 3. Only cluster candidates that have more than *Pts^min^* members of the same property are recognized as possible clusters in the next FCM quantization process. Pixels which belong to clusters that are not qualified (no more than *Pts^min^* members) are removed as possible noise.

Then, the DBSCAN process used in this paper is summarized in Algorithm 1.
**Algorithm** **1.** DBSCAN process.[Step 1]Initialize *Pts^min^* = 3, *ε* = 1, DBSCAN_ct_ = 0 and density d=(W×H)/255 where *W* and *H* denote the width and the height of the region of interest.[Step 2]Let *Da* be the frequency of intensity value as the number of pixels. For every k-th intensity value, perform step 3 and step 4 until reaching the highest intensity value in the region of interest.[Step 3]Among the high intensity area from histogram analysis, make clusters within radius ε with consecutive intensity values larger than d; if the cluster satisfies *Pts^min^* threshold for that cluster, increase DBSCAN_ct_ by 1. Otherwise, remove all pixels in that candidate cluster.[Step 4]Among the low intensity area from the histogram analysis, make clusters within radius *ε* with consecutive intensity values no larger than d; if the cluster satisfies the *Pts^min^* threshold, increase DBSCAN_st_ by 1. OItherwise, remove all pixels in that candidate cluster.

The variable DBSCAN_ct_ represents the number of clusters DBSCAN finds as the candidate of the ganglion cyst.

Then, the number of clusters that will be used in FCM quantization process is computed by the Equation (3).
(3)$$\begin{array}{c} if\left(DBSCAN_{pmax}\leq \frac{\sum_{h=1}^{W}\sum_{g=1}^{H}P_{gh}\left(x,y\right)}{W\times H}or\;DBSCAN_{ct}\geq 4\right)\;Fcm_{ct}=0\\ else\;if\left(\;DBSCAN_{ct}=1\right)\;Fcm_{ct}=4\;\\ else\;Fcm_{ct}=2\; \end{array}$$

*FCM* will not be applied when the maximum intensity value among pixels in the same cluster, *DBSCAN*_pmax_ is smaller than the average intensity or the number of clusters satisfying minimum density condition (*DBSCAN_ct_*) is 4 or more, DBSCAN is considered to have effective low intensity clusters thus FCM is not required (*FCM_ct_* = 0). Otherwise, if DBSCAN has only one cluster, our method set the number of clusters parameter for *FCM*(*FCM_ct_*) = 4 else *FCM_ct_* = 2 and backend FCM quantization process begins.

Three cases shown in Figure 4 explain Equation (3) with respect to the DBSCAN quantification. In case of image 1 (Figure 4a), the number of clusters DBSCAN found was 5, which is enough to locate the cyst area as shown in Figure 4d. For image 2 of Figure 4b, DBSCAN_pmax_ was smaller than average intensity value (i.e., the target area is sufficiently contrasted from the background). Thus, FCM will not be applied in those two cases. However, DBSCAN finds only one cluster from Figure 4c as shown in Figure 4f. For this case, FCM needs enough clusters to extract the target area from background. Cases other than shown in Figure 4, we set the number of clusters parameter equal to 2 for FCM quantification process.

### 2.3. FCM as the Backend Segmentation Process

FCM assigns the membership value of every pixel of an image based on the distance measure (Euclidean distance. m = 2 in this paper) to the centroid of each cluster. The cluster centroids and the corresponding membership values are iteratively updated as the members of the same cluster have more similarity. We apply FCM quantification process after DBSCAN and the FCM process in this paper can be summarized as shown in Algorithm 2.
**Algorithm 2.** FCM quantification process.[Step 1]Initialize *c* (2 ≤ *c* < *n*) for n pixels of the area as the result of (3) by DBSCAN quantization, and exponential weight m (1 ≤ *m* < ∞). Also initialize the error threshold (*ε*) for terminating condition and the membership degree *U*(0).[Step 2]Compute the centroid of a cluster as the mean of all points, weighted by their degree of belonging to the cluster as following, where *i* denotes the cluster number, *j* denotes the node number on input *x* of total *n* data. $$v_{ij}=\overset{n}{\underset{k=1}{\sum }}{\left(u_{ik}\right)}^{m}x_{kj}/\overset{n}{\underset{k=1}{\sum }}{\left(u_{ik}\right)}^{m}$$[Step 3]Then, compute the distance between the data point and each centroid point of the cluster as following where *i* denotes the number of nodes. $$d_{ik}={\left[\overset{l}{\underset{j=1}{\sum }}{\left(x_{kj}-v_{ij}\right)}^{2}\right]}^{1/2}$$[Step 4]Then, update the membership function *U* of its (*r* + 1)th repetition as following.
$$u_{ik}^{\left(r+1\right)}=\frac{1}{\sum_{j=1}^{c}{\left[\frac{d_{jk}^{r}}{d_{ik}^{r}}\right]}^{2}}\;for\;I_{k}=\emptyset$$[Step 5]Repeat above steps until the difference between *U*^r+1^ and *U*^r^ becomes less than predetermined threshold value.

The overall flow of the proposed method including DBSCAN can be summarized as shown in Figure 5.

An example for the whole ganglion cyst extraction is illustrated in Figure 6. The input image Figure 6a was quantified by DBSCAN, but the result showed there was only 1 cluster candidate (Figure 6b) thus we set the number of clusters c = 4 for FCM as Equation (3) and perform FCM quantification as shown in Figure 6c. Then, noise is filtered by fuzzy stretching as explained in Section 2.1 (Figure 6d). Finally, the object formation process is applied to extract the target area as shown in Figure 6e.

## 3. Results

### 3.1. Environment

The proposed method is implemented with C# under Microsoft Visual Studio 2019 on an IBM-compatible PC with AMD Ryzen 3 3300X 4-Core Processor 3.70 GHz, 16GB RAM. The experiment uses 120 DICOM format ultrasound images that contain the wrist ganglion cyst are provided from Gupo Sungsim Hospital, Busan, Korea. Ultrasonography is performed by Philips iU22 using 3–5 MHz transducer. Two radiologists from that hospital verified the system’s ganglion cyst extraction results compared with their own hand-picked cyst area extraction as the ground truth of this experiment.

### 3.2. Performance Measure and Experimental Result

In this paper, the proposed method is compared with standalone DBSCAN and standalone FCM and the accuracy, recall (sensitivity), precision and *F*1 score were employed as performance indices, as shown in Equation (4).
(4)$$\begin{array}{c} accuracy=\frac{TP}{TP+FN+FP}\times 100\\ Recall=\frac{TP}{TP+FN}\times 100\\ \begin{array}{c} Precision=\frac{TP}{TP+FP}\times 100\\ F1\;Score=2\left(\frac{Precision\times Recall}{Precision+Recall}\right) \end{array} \end{array}$$

Since the ground truth is the area that human radiologist marked on the same image, TP in Equation (4) represents the number of pixels in the true positive area (i.e., pixels in the human picked cyst area), FP represents the number of pixels in the false positive area (i.e., pixels the software recognized as the part of the cyst but not included in the ground truth area), and FN represents the number of pixels in the false negative area (i.e., the software did not recognize the ground truth). Since the ground truth was human expert’s choice of cyst area, the true negative area (both computer software and human expert did not choose a pixel as the part of the ganglion cyst) was meaningless. Thus, it was not included in the performance indices.

The overall performance result is summarized in Table 1.

Firstly, human radiologists verify if the location of the extracted ganglion cyst from the software (DBSCAN only, FCM only, and proposed method) is congruent to their own results. Standalone DBSCAN greatly suffers from fuzzy overlapped pixels from the given images as only 31.7% of its extractions are accepted by human expert. Scores of other performance indices are computed only from images that are correctly extracted images. The proposed method was the best among three methods in that 89.2% of its ganglion cyst extractions are accepted by human experts.

Among those accepted ganglion cyst extractions, it is interesting that the proposed method is statistically better than FCM alone except the recall index while the precision of FCM is even worse than that of DBSCAN. From Equation (4), we can infer that FCM tends to have more FP and less FN than the proposed method. That means FCM tends to recognize wider area than the proposed method in segmentation.

Among 120 tested sonographic images, 81 images are accepted to designate correct ganglion cyst location by human expert with both FCM and the proposed method. We compare how two methods behave as the number of pixels both methods found in the TP, FP, and FN areas (as shown in Table 2).

The average number of pixels in ground truth among 81 tested images are 6958 and we can clearly see the difference of two methods. The proposed method is more conservative but more congruent to human expert.

The size of ganglion cysts (ground truth) among tested 120 images varies from 220 pixels to 31,514 pixels. Knowing that larger ganglion cysts more likely had well-defined borders [6] and thus easier to extract by computer algorithms, we will take another analysis based on the size of ground truth pixels of the images. The median size of the ground truth ganglion cysts among 120 tested images is 4647. Thus, we divided two groups of images that have larger cysts and smaller cysts (both have 60 images). Then, we take the same performance comparison among three methods in Table 3.

Although among large cysts group images the proposed method and FCM behaves differently, the accuracy is not statistically different. However, among the smaller cysts image group, the proposed method shows far better accuracy and F1 score and FCM takes more false positive pixels in its extracted ganglion cyst area thus we may conclude that the proposed method is quantitatively and qualitatively better than FCM alone for the wrist ganglion cysts extraction problem.

## 4. Conclusions

In this paper, we propose an automatic segmentation of wrist ganglion cysts from ultrasonographic images for reducing the examiner subjectivity effect which frequently arises in manual ultrasonography analysis. We used two artificial intelligence algorithms, namely DBSCAN and FCM. The proposed method uses DBSCAN as the front end and FCM as the back end in this segmentation problem. The advantage of DBSCAN is the finding of the natural arbitrary shape of the cluster. Thus, it gives some a priori information to the back-end segmentation process by FCM. With that a priori information earned from DBSCAN quantification, the number of clusters used in FCM is determined from 0 to 4 by the Equation (3).

In an experiment with 120 ultrasonographic images containing wrist ganglion cysts of various sizes and shapes, the proposed method behaves much like human radiologist does. The proposed method successfully locates the cyst area in 107 of 120 cases (89.2%) as opposed to 85 cases of standalone FCM and 38 cases of standalone DBSCAN. Also, we found that the proposed method shows lower false positive rate when the ground truth is set to the human radiologist’s choice of the ganglion cyst area. Such a trend is more apparent when small size ganglion cyst is given in the input image. The proposed method is significantly statistically better in terms of accuracy, precision, and F1 score than FCM. Since it is known that small size ganglion cysts are often hypoechoic and hard to find accurately thus the advantage that the proposed method has in small size cyst images is meaningful.

While the proposed method is fully automatic in that no human intervention is required in analyzing input sonography, there are several places that this research can be extended to in the future. Using a density-based algorithm such as DBSCAN gives good priority for decide the number of clusters in FCM quantization process with its ability to make shape independent cluster. However, the performance of DBSCAN alone is too low in extracting the target area (ganglion cyst) thus, we may need a better method in using density information other than standard DBSCAN we used in this paper. In another direction, with retrospective analysis of failed segmentation cases by our proposed method, our method is weak when the distribution of intensity in histogram analysis is scattered within a small range. In such cases, our method tends to recognize a greater number of clusters than required thus it picked different location compared with human experts. To mitigate this effect, we need more careful noise reduction process before quantization process or considering spatial information in FCM quantization process.

## Figures and Tables

**Figure 1 diagnostics-11-02329-f001:**
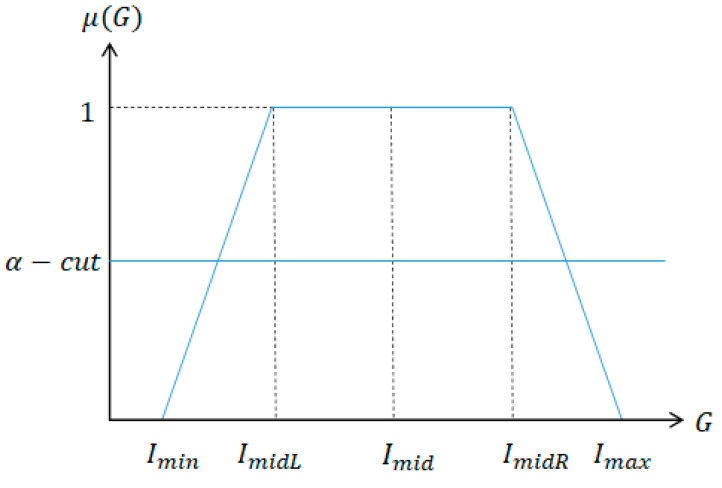
Trapezoidal membership function.

**Figure 2 diagnostics-11-02329-f002:**
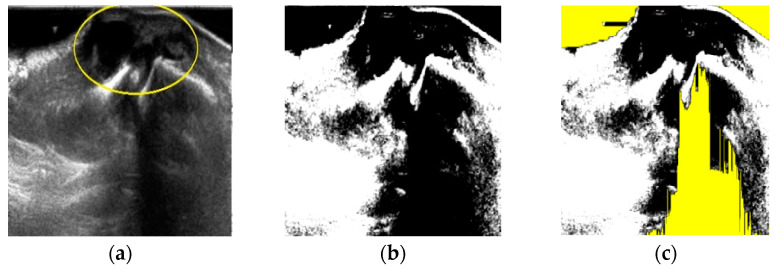
The effect of fuzzy stretching and noise reduction. (**a**) Input image, (**b**) after stretching, (**c**) noise reduction (yellow).

**Figure 3 diagnostics-11-02329-f003:**
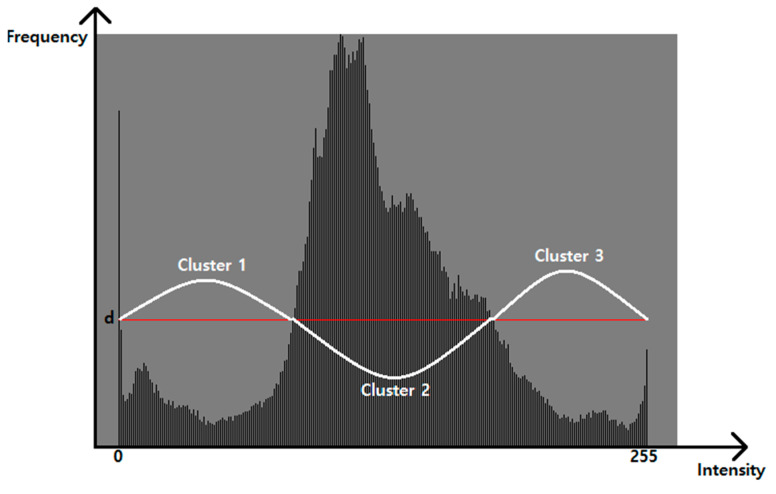
Histogram analysis for DBSCAN algorithm.

**Figure 4 diagnostics-11-02329-f004:**
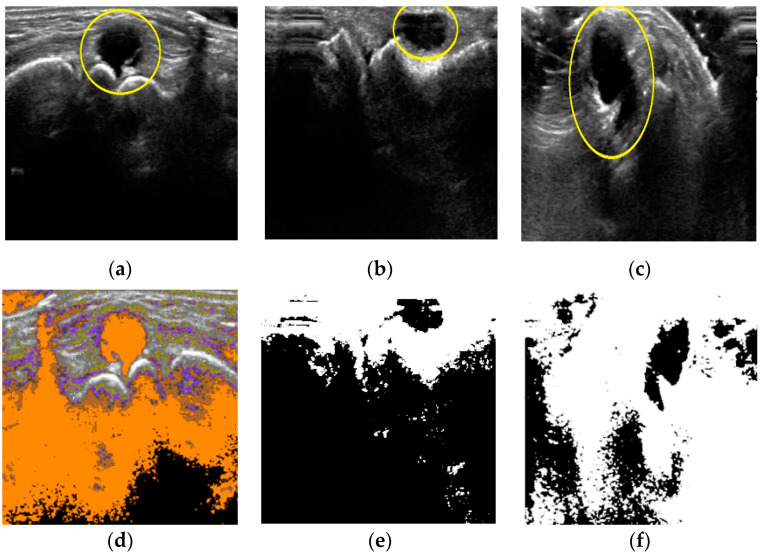
Cases of DBSCAN quantifications. (**a**) Input image 1; (**b**) input image 2; (**c**) input image 3; (**d**) #-of-clusters = 5; (**e**) dark cluster; (**f**) one cluster.

**Figure 5 diagnostics-11-02329-f005:**
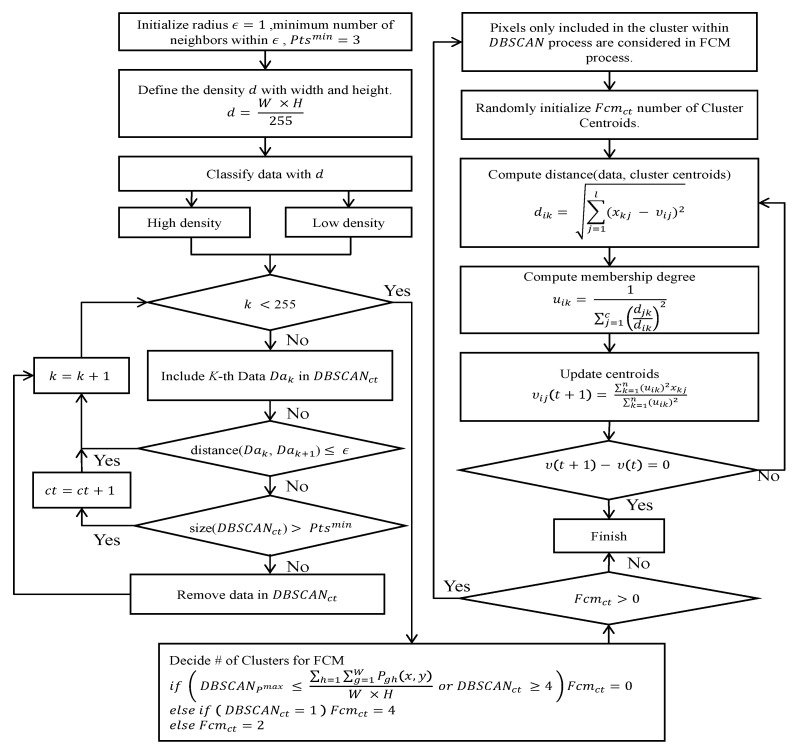
Overall clustering and quantization process.

**Figure 6 diagnostics-11-02329-f006:**
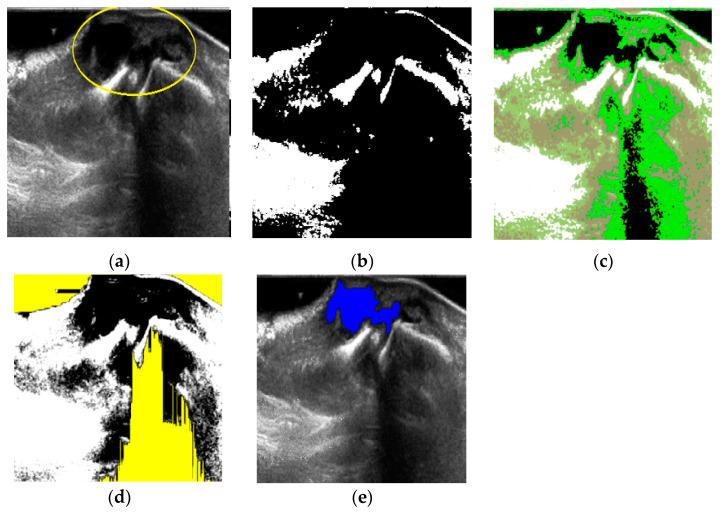
Ganglion cyst extraction process. (**a**) Input image 1; (**b**) *Cluster* = 1 after DBSCAN; (**c**) FCM quantification; (**d**) noise by fuzzy stretching; (**e**) extraction by labeling.

**Table 1 diagnostics-11-02329-t001:** Overall performance results (120 cases). Colored, statistically significant (*p* < 0.05).

Method	Accuracy	Recall	Precision	*F*1 Score	Extractions	Ext. Rate
DBSCAN	57.26%	71.80%	82.21%	69.46%	38	31.7%
FCM	72,19%	89.35%	80.47%	82.74%	85	70.8%
Proposed	75.43%	81.34%	91.91%	84.44%	107	89.2%

**Table 2 diagnostics-11-02329-t002:** Extracted area in average # of pixels (81 cases).

Method	Truth	TP	FP	FN
FCM	6958	6370	1358	588
Proposed	6958	5553	355	1406

**Table 3 diagnostics-11-02329-t003:** Performance result with respect to the size of the ganglion cyst. Colored, statistically significant (*p* < 0.05).

Size	Method	Accuracy	Recall	Precision	*F*1 Score	Cases
Large	DBSCAN	57.37%	62.98%	92.96%	59.46%	16
	FCM	77.97%	90.37%	85.38%	87.13%	45
	Proposed	78.40%	83.00%	93.96%	86.83%	58
Small	DBSCAN	57.07%	78.22%	74.38%	70.12%	22
	FCM	65.69%	88.20%	74.95%	77.80%	40
	Proposed	71.92%	78.37%	89.48%	81.62%	49

## Data Availability

The data presented in this study are available on request from the corresponding author. The data are not publicly available due to Institutional regulations.

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
