# Peer review of "Intelligent Automatic Segmentation of Wrist Ganglion Cysts Using DBSCAN and Fuzzy C-Means"

_diagnostics, 2021, doi:10.3390/diagnostics11122329_

Round 1

Reviewer 1 Report

>The introduction should contain additional information (with references) related to other solutions applied to similar issues (medical problems).

>According to my suggestion, the following information (the first section – ‘Introduction’) about data clustering and images analysis is needed (used methods, software implementations, etc.).

>Please improve the description of the symbols used in the formulas.

>I suggest presentation of the algorithms as the schematic diagrams (page 4, page 5).

>Details of the software implementation of the applied techniques are needed.

>How was the radius of the DBSCAN selected?

>What is the influence of the above-mentioned parameter on the final results?

>How were the clustering methods initialized? Does this process affect the results of comparison?

>Have you considered the disturbances in the original image?

>How were the triggers (switching values) determined (equation 3)?

Reviewer 2 Report

I tried to analyse your paper for some errors, mistakes, missing parts or uncomplete ideas.. But I found only several issues. Reading your paper was inspirative for me!

What i am missing in the paper, is the more clear reason for this research. More clear defition of problem you would like to solve....

And also just summary what I need to be in article and what already is inside..:

- Abstract contain conclusions with values.. == OK, and also values.. maybe only some more exact contribution highlight..

- graphs/tables from results with values, X and Y axis labels...== OK.

- flow chart of solution / proces / architecure... NOT ..  so please add this piece to articel.. especialy for Diagnostics it is mportant.. 

- references not only in introduction but in wohle article while contribution is based on them.. == not at all parts.. some update is suggested. if any statement is added, you need to refer to existing literature..

- references to this journal - as to prove closenes of the topic == NOT - need to be added some.. .. but also other IEEE trans. references need to be used.. .

- some not so long conclusions - including numbers, values, pros and cons.. == OK. also with values.. 

- future directions in the end of conclusion.. == missing/not clear

- reasonable number of references.. == can be added more referencs.. as i mentioned.. at leat 30 in total.

- Q1/Q2 journal articles in references.. == some need to be added

So please update as MINOR revisions from me!

Author Response

Reviewer #2

Thank you for your valuable detailed comments to improve our paper.

Comments and Suggestions for Authors

I tried to analyse your paper for some errors, mistakes, missing parts or uncomplete ideas.. But I found only several issues. Reading your paper was inspirative for me!

What i am missing in the paper, is the more clear reason for this research. More clear defition of problem you would like to solve....

And also just summary what I need to be in article and what already is inside..:

  1. Abstract contain conclusions with values.. == OK, and also values.. maybe only some more exact contribution highlight..

→ We make our contribution clearer in the abstract and conclusion.

Such human-like behavior is more apparent when the size of ganglion cyst is small where the quality of ultrasonography is often not very high. With this fully automatic segmentation method, the operator subjectivity that is highly dependent on the experience of ultrasound examiner can be mitigated with high reliability.

  1. graphs/tables from results with values, X and Y axis labels...== OK.

- flow chart of solution / proces / architecure... NOT ..  so please add this piece to articel.. especialy for Diagnostics it is mportant.. 

→ We add flow chart/detailed explanation on two major algorithms we used and brief explanation on the preprocessing part that is done before the theme of this paper.

  1. References

- references not only in introduction but in wohle article while contribution is based on them.. == not at all parts.. some update is suggested. if any statement is added, you need to refer to existing literature..

- references to this journal - as to prove closenes of the topic == NOT - need to be added some.. .. but also other IEEE trans. references need to be used.. .

- reasonable number of references.. == can be added more referencs.. as i mentioned.. at leat 30 in total.

- Q1/Q2 journal articles in references.. == some need to be added

→ We added several more references as following;

  1. Sultan, E.; Ahmad, N.; Daryoush, A.S. Diagnosis of hand ganglion cyst using free space broadband frequency modulated fNIR imaging system. In 2014 IEEE Benjamin Franklin Symposium on Microwave and Antenna Sub-systems for Radar, Telecommunications, and Biomedical Applications 2014, 1-3.
  2. Gopinath, K.; Sivaswamy, J. Segmentation of retinal cysts from optical coherence tomography volumes via selective enhancement. IEEE journal of biomedical and health informatics 2018, 23, 273-282.

  1. Huang, H.; Meng, F.; Zhou, S.; Jiang, F.; Manogaran, G. Brain image segmentation based on FCM clustering algorithm and rough set. IEEE Access 2019, 7, 12386-12396.
  2. Kim, K.B.; Song, D.H. Intelligent automatic extraction of canine cataract object with dynamic controlled fuzzy C-means based quantization. International Journal of Electrical and Computer Engineering 2018, 8, 666-672.
  3. Rehman, S.N.; Hussain, M.A. Fuzzy C-means algorithm-based satellite image segmentation. Indonesian Journal of Electrical Engineering and Computer Science 2018, 9, 332-334.
  4. Zhou, J.; Wang, J.; Bu, H. Fabric defect detection using a hybrid and complementary fractal feature vector and FCM-based novelty detector. Fibres & Textiles Eastern Europe 2017, 25, 46–52.
  5. Mohammed, K.M.C.; Kumar, S.S.; Prasad, G. Optimized Fuzzy C-means Clustering Methods for Defect Detection on Leather Surface. Journal of Scientific and Industrial Research 2020, 79, 833-836.

  1. Lei, T.; Jia, X.; Zhang, Y.; Liu, S.; Meng, H.; Nandi, A.K. Superpixel-based fast fuzzy C-means clustering for color image segmentation. IEEE Transactions on Fuzzy Systems 2018, 27, 1753-1766.

  1. Conclusions

- some not so long conclusions - including numbers, values, pros and cons.. == OK. also with values.. 

- future directions in the end of conclusion.. == missing/not clear

→ We add future research direction in the conclusion part.

While the proposed method is fully automatic in that no human intervention is required in analyzing input sonography, there are several places that this research can be extended in the future. Using density-based algorithm like DBSCAN gives good priori to decide the number of clusters in FCM quantization process with its ability to make shape independent cluster. However, the performance of DBSCAN alone is too low in extracting the target area (ganglion cyst) thus, we may need a better method in using density information other than standard DBSCAN we used in this paper. In another direction, with retrospective analysis of failed segmentation cases for our proposed method, our method is weak when the distribution of intensity in histogram analysis is scattered within a small range. In such cases, our method tends to recognize a greater number of clusters than required thus picked different location compared with human experts. To mitigate this effect, we need more careful noise reduction process before quantization process or considering spatial information in FCM quantization process.